# Antiferromagnetism and Structure of $Sr_{1-x}Ba_xFeO_2F$ Oxyfluoride Perovskites

**Crisanto A. Garcia-Ramos** [1,2,*] , **Kiril Krezhov** [2,3,*] , **María T. Fernández-Díaz** [4] and **José A. Alonso** [1]

1 Instituto de Ciencia de Materiales de Madrid, CSIC, Cantoblanco, E-28049 Madrid, Spain
2 Institute for Nuclear Research and Nuclear Energy, Bulgarian Academy of Sciences, Tsarigradsko Chaussee 72, BG-1784 Sofía, Bulgaria
3 Institute of Electronics, Bulgarian Academy of Sciences, Tsarigradsko Chaussee 72, BG-1784 Sofía, Bulgaria
4 Institute Laue Langevin, BP 156X, F-38042 Grenoble, France
* Correspondence: crisanto_garcia@yahoo.com (C.A.G.-R.); kiril.krezhov@gmail.com (K.K.)

**Abstract:** Recently, a series of oxyfluorides, $Sr_{1-x}Ba_xFeO_2F$ with x = 0, 0.25, 0.50, and 0.75 obtained through a novel synthesis route, were characterized by X-ray and neutron powder diffraction (NPD), magnetization measurements, and $^{57}Fe$ Mössbauer spectroscopy (MS). The diffraction data revealed random occupancy of Sr and Ba atoms at the A-cation site, and a statistical distribution of O and F at the anionic sublattice of the perovskite-like structure specified in space group Pm-3m. MS spectra analysis consistently indicated the presence of $Fe^{3+}$ ions at B-site, confirming the $Sr_{1-x}Ba_xFeO_2F$ stoichiometry. Magnetic structure determination from the NPD data at room temperature established G-type antiferromagnetic arrangement in all compositions with $Fe^{3+}$ moments of about 3.5 µB oriented along the c axis. In this study, we present and analyze additional NPD data concerning the low-temperature chemical and magnetic structure of $Sr_{0.5}Ba_{0.5}FeO_2F$ (x = 0.5) and $SrFeO_2F$ (x = 0). Basically, the three-dimensional G-type magnetic structure is maintained down to 2 K, where it is fully developed with an ordered magnetic moment of 4.25(5) µB/Fe at this temperature for x = 0.5 and 4.14(3) µB/Fe for x = 0. The data processing is complemented with a new approach to analyze the temperature dependence of the magnetic order $T_N$ on the lattice parameters, based on the magnetic hyperfine fields extracted from the temperature-dependent MS data.

**Keywords:** magnetic interaction energy; hyperbolical tangent; Hamiltonian; neutron powder diffraction (NPD); Mössbauer spectroscopy (MS); ferromagnetic (FM); antiferromagnetic coupling (AFM)

## 1. Introduction

Recently, several papers have addressed the synthesis and properties of oxyfluoride perovskites $Sr_{1-x}Ba_xMO_2F$ (M = transition metal) [1–6]. This type of perovskite exhibits a mixed anion sublattice, which incorporates both oxygen and fluoride atoms, and may display appealing properties. The insertion of fluoride ions into transition metal oxides leads to a change in the transition metals' oxidation states, which involves the intention of modifying their electronic structure and thus changing their magnetic and electrical properties. In fact, they have attracted a lot of attention since the discovery of superconductivity in cuprate oxyfluorides of $Sr_2CuO_2F_{2+x}$ stoichiometry [6].

Since direct solid-state reactions may only yield thermodynamically stable simple oxyfluorides, topotactic reactions from suitable oxide precursors with the adequate crystal structure are preferred for fluorination. These involve low-temperature treatments with appropriate fluorinating agents. For this reason, low-temperature reagents such as $F_2$ gas, $NH_4F$, $MF_2$ (M = Cu, Ni, Zn), or $XeF_2$ are usually utilized. Some sophisticated synthetic pathways of several stages, such as the preparation of $SrFeO_2F$ starting from $SrFeO_{3-\delta}$, have been published [7,8].

A new, simpler synthesis procedure implying the treatment of oxide precursors with a F-containing polymer (polyvinylidene fluoride) has been implemented, and it

was successful for the synthesis of such oxyfluorides as $SrFeO_2F$, $BaFeO_2F$, $Ca_2CuO_2F_2$, and $Sr_2TiO_3F_2$ [9–12]. The compounds of the series $Sr_{1-x}Ba_xFeO_2F$ were prepared and studied by X-ray diffraction (XRD) by Clemens et al. [13]. In their pioneering work, they described these oxyfluorides as cubic perovskites, independently of the concentration, $x$, defined in the space group Pm-3m [13]. Later on, $BaFeO_2F$, $Sr_{0.5}Ba_{0.5}FeO_2F$ and $SrFeO_2F$ were studied by $^{57}Fe$ Mössbauer spectroscopy (MS), determining that a single $Fe^{3+}$ valence is present in the octahedral sites, as was also verified from neutron powder diffraction (NPD) data, especially for the compound $BaFeO_2F$ [9,10,14,15]. Indeed, the problem of the formation and the physical properties of oxyfluoride perovskites is one of the most important in condensed matter physics, and a significant number of publications have been devoted to it in the last years [12–15].

In this paper, we report on complementary data for the oxides of nominal composition $Sr_{1-x}Ba_xFeO_2F$ (x = 0, 0.25, 0.5, 0.75) regarding low-temperature NPD data that permitted the resolution of the magnetic structures for the perovskites with x = 0 and x = 0.50. We additionally analyzed the temperature dependence of the magnetic order $T_N$ on the lattice parameters, based on the magnetic hyperfine fields extracted from the temperature-dependent MS data.

## 2. Research Methodology

### 2.1. Structure and Magnetic Interaction Energy of Oxyfluoride Perovskites

Neutron powder diffraction and Mössbauer data analysis of the perovskite series $Sr_{1-x}Ba_xFeO_2F$ showed a cubic structure with the group Pm-3m. A view of the crystal structure is illustrated in Figure 1 [5,13]. In this family, the $a$ lattice parameter can be expressed as:

$$a(x) = \sum_{j=1}^{2} \sqrt[3]{\frac{V_j}{Z_j}}.x_j \qquad (1)$$

which depends on the concentration $x_j$ (Ba/Sr) of barium/strontium elements, and it expresses the lattice parameter $a$ as a function of $x$; $V_j$, $Z_j$—volume and atomic number of elements. For the perovskite series $Sr_{1-x}Ba_xFeO_2F$, the expression for the lattice parameter takes the form:

$$a_x = x_{Sr} (V_{Sr}/Z_{Sr})^{1/3} + x_{Ba} (V_{Ba}/Z_{Ba})^{1/3}, \text{ yielding}$$
$$a_x = (1 - x) a_{Sr} + x. a_{Ba} \qquad (2)$$

where $x$ stands for the content of Ba, and $a_{Sr}$ and $a_{Ba}$ are the ionic sizes of elements $Sr^{2+}$ and $Ba^{2+}$ cations.

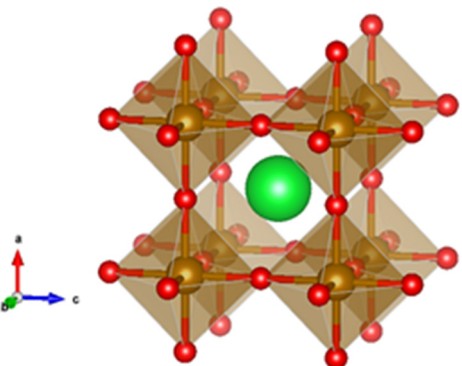

**Figure 1.** View of the cubic crystal structure for $Sr_{1-x}Ba_xFeO_2F$ perovskites, with the voluminous Sr and Ba atoms (green sphere) distributed at random at the center of the cube determined by Fe atoms (brown) in octahedral coordination with oxygen and fluor (red spheres).

The Hamiltonian for the Fe atom in the perovskite series $Sr_{1-x}Ba_xFeO_2F$ is usually composed by electronic Zeeman interaction, the exchange between the spin of four Fe

atoms, hyperfine interactions between nuclear spin and electronic spin, nuclear Zeeman interaction, and the nuclear quadrupole interaction. It has the form:

$$H = \mu_B \sum_j H.g.S + \frac{1}{2} \sum_{i,j;i\neq j} J_{ij}.S_i.S_j. + \sum_i S_i.A_i.I_i - \mu_N \sum_i g_n.H.J_i + e^2 \sum_i Q \frac{(V_{zz})_i}{4I(2I-1)} \left[ 3I_{zi}^2 - I^2 + \frac{n_i}{2} \left( I_+^2 - I_-^2 \right) \right] \quad (3)$$

where $S_i$, $I_i$, $J_{ij}$, $A_i$, $V_{zz}$, $n_i$, $\mu_B$, $\mu_N$, $g_i$, $g_n$ and $e$ are the electronic spin, nuclear spin of iron atoms; the exchange coupling between electronic spin; magnetic hyperfine coupling; the nuclear quadrupole moment; gradient electrical field; Bohr magneton; nuclear magneton; and electronic g-factor, nuclear g-factor and electronic charge, respectively. The Mössbauer data showed that the magnetic interaction energy is 340 times greater than electrical quadrupole energy in the sextet: $\xi$(sextet at 77 K)$/\xi$(quadrupole splitting) = (17.38 mms$^{-1}$/0.05 mms$^{-1}$) $\approx$ 347, so we focus only on the magnetic hyperfine interaction between the Fe atoms.

The interaction magnetic energy between two atoms Fe is usually expressed [16,17] as:

$$\xi = \frac{\mu_1 \mu_{2-3(\mu_{1.}\ e)(\mu_{2.}\ e)}}{a^3} . \frac{\mu_m}{4\pi}, \ \mu_i = g.\mu_B.S_i, \ i = 1,\ 2$$

where $\mu_{i.}$, $S_i$—magnetic moment and spin of each atom Fe, $\mu_m$—magnetic permeability of medium, $\mu_B$—Bohr's magneton, $e = \frac{\{x,y,z\}}{(x^2+y^2+z^2)^{3/2}}$—unity vector between two atoms Fe, one of them in position {0, 0, 0} and the other in position {x, y, z}.

In the members of the $Sr_{1-x}Ba_xFeO_2F$ perovskite series (antiferromagnetic below $T_N$), it is necessary to calculate all the contributions of each atom in the neighborhood of a given central atom. For this purpose, assuming that the central atom is in position {0,0,0}, we will have a magnetic moment $\mu_1$ along the positive z-axis ($\uparrow$), the second atom in position {x, y, z} will have moment $\mu_2${0, 0, $(-1)^{x+y+z}$} oriented positively on the direction of the z-axis ($\uparrow$) if the sum (x + y + z) is even, or negative direction of the z-axis ($\downarrow$) if the sum is odd. The interaction energy can be expressed as [18–20]:

$$\xi = \frac{\mu_m}{4\pi} \sum_{-N}^{N} \sum_{-N}^{N} \sum_{-N}^{N} \frac{\mu_1 \mu_2 (-1)^{x+y+z} - 3\mu_1\mu_2 \frac{(-1)^{x+y+z} xz^2}{x^2+y^2+z^2}}{a^3 (x^2 + y^2 + z^2)^{3/2}} \quad (4)$$

where $N$ indicates the number of atoms with coordinates {x, y, z} in the vicinity of the central atom in position {0, 0, 0}. In this type of perovskite, we have identified the following magnetic couplings: antiferromagnetic coupling (AFM) and ferromagnetic coupling (FM) for N = 1, as shown in Figure 2, where $J_1$ and $J_5$ are AFM, $J_2$ and $J_3$ are FM, and the contribution of $J_4$ is AFM is zero due to its position vector {a, a, a} with its unit vector $e = \{1/\sqrt{3}, 1/\sqrt{3}, 1/\sqrt{3}\}$, giving a null interaction energy in Equation (4).

$$\xi = [-\mu_1.\mu_2 + 3(\mu_1.e)(\mu_2.e)]/a^3 = [-\mu_1.\mu_2 - 3\ \mu_1.\mu_2\ (\frac{1}{\sqrt{3}})(-\frac{1}{\sqrt{3}})]/a^3 = 0$$

This hyperfine interaction energy will be equal to the product ($K_B.T_N$) due to molecular vibration by heat, where $K_B$ is the Boltzmann constant and $T_N$ the Néel temperature:

$$K_B.\ T_N = \beta.\ \frac{\mu_1\ \mu_2}{a^3} \quad (4a)$$

where the $\beta$ parameter of proportionality can be evaluated using data obtained by MS data of the perovskite $SrFeO_2F$.

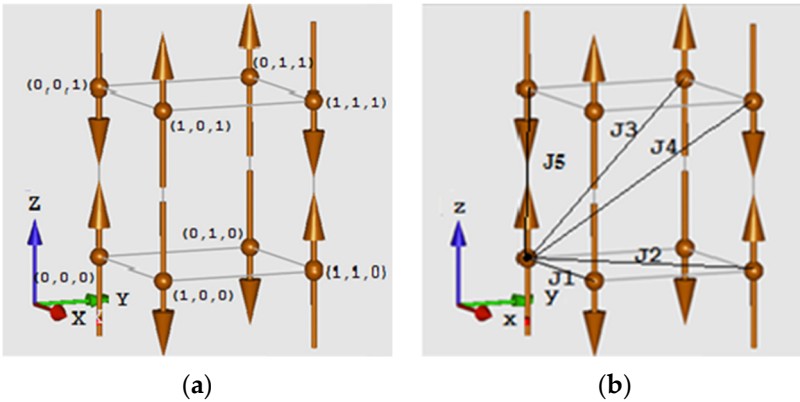

**(a)**                                                         **(b)**

**Figure 2.** (**a**) View of the structure of the $Sr_{1-x}Ba_xFeO_2F$ perovskite and the magnetic moments of the atoms aligned on the z-axis upwards (↑) or downwards (↓) according to their position {x, y, z}; (**b**) AFM ($J_1$, $J_4$, $J_5$)/FM ($J_2$, $J_3$) interaction between the magnetic moments of atoms.

## 2.2. Experimental Section

### 2.2.1. NPD Data Acquisition

Low-temperature NPD patterns were collected for $Sr_{1-x}Ba_xFeO_2F$ (x = 0, 0.5) at the high-flux D20 neutron diffractometer of the Institut Laue Langevin (Grenoble-France), coupled with a standard orange cryostat. The samples were contained in vanadium cans. A wavelength of 2.40 Å was selected from a graphite monochromator. Good statistical patterns were collected at the lowest temperature (2 K) for 1 h, then a sequential collection was launched in the 2–160 K interval, with step of 20 K and an acquisition time of 10 min for each diagram. The patterns were refined by the Rietveld method [21] using the *Fullprof* refinement program [22]. A pseudo-Voigt function was chosen to generate the line shape of the diffraction peaks. No regions were excluded in the refinement. In the final run, the following parameters were refined: scale factor, background coefficients, zero-point error, unit-cell parameters, pseudo-Voigt corrected for asymmetry parameters, positional coordinates, and isotropic displacement factors. For the G-type magnetic structure, the magnitude of the Fe magnetic moment as also refined. The coherent scattering lengths for Sr, Ba, Fe, O and F atoms were 7.020, 5.079, 9.45, 5.803 and 5.654 *fm*, respectively.

### 2.2.2. Mössbauer Spectroscopy

The Mössbauer spectrum was collected utilizing a conventional spectrometer with a [57]Co/Rh source, in transmission mode. The signal-to-noise ratio was optimized, avoiding saturation with a sample thickness of 10 mg of natural Fe/$cm^2$. The spectra were analyzed by means of a nonlinear fit, and the MIF-Mössbauer integral fit program was used, based on the approximation of an integral Lorentzian line shape [23–25]. The high-temperature measurements were carried out in an NB sample holder, whereas for the low-temperature range, a second one with Be windows was used. For the $Sr_{1-x}Ba_xFeO_2F$ (x = 0.00, 0.25, 0.50, 0.75) oxides, the [57]Fe Mössbauer spectra were recorded at 77 K using liquid nitrogen, 300 K (room temperature), and, in a specially designed furnace, at 373 K, 473 K, 573 K, 673 K, 723 K, 823 K, 873 K and 923 K. The isomer shifts (IS) of the spectra refer to the centroid of an α-fe foil (6 μm) reference spectrum at room temperature (RT) [23].

## 3. Results and Discussion

Neutron diffraction experiments (NPD) at RT were described in a previous publication [1], assessing that all the compounds of the $Sr_{1-x}Ba_xFeO_2F$ series crystallize with cubic symmetry, defined in the Pm-3m space group. At RT, NPD data already show the presence of a magnetic structure well known as G-type, since the Néel temperature ($T_N$) of these compounds is well superior to RT. In the present study, we aimed at investigating the low-temperature crystal and magnetic structures, in order to examine the totally consolidated magnetic array, as well as to check the persistence of the cubic structural arrangement at 2 K.

It is well known that the determination of the O positions in oxide networks is difficult by X-ray diffraction, given the weak scattering factor for $O^{2-}$ ions; hence, neutron diffraction measurements are essential. It is also known that many $ABO_3$ perovskite oxides that are cubic at RT may experience phase transitions at lower temperatures, with a reduction in symmetry due to the tilting of $BO_6$ octahedra, giving rise to superstructure peaks in the diffraction patterns.

The low-temperature crystal structures of $Sr_{1-x}Ba_xFeO_2F$ (x = 0, 0.5) could indeed be refined in the cubic Pm-3m space group down to 2 K. There were no symptoms of any reduction in symmetry down to the lowest temperature for the two studied compositions. In the cubic model, Sr and Ba are statistically distributed at the 1*b* Wyckoff sites (¹/₂ , ¹/₂ , ¹/₂); Fe is located at 1a positions (0, 0, 0), whereas there is a unique oxygen and fluorine, distributed at random at 3*c* (¹/₂ , 0, 0) sites. As O and F exhibit very similar scattering lengths, their relative occupancy could not be refined. Thus, the average structure observed by NPD is cubic since the long-range order of the anion displacements, if any, should be suppressed by the structural disorder arising from the random distribution of oxide and fluoride ions. Figure 3a,b illustrate the goodness of the fit for the NPD data for $SrFeO_2F$ and $Sr_{0.5}Ba_{0.5}FeO_2F$, respectively. The following unit-cell parameters were refined at 2 K: a = 3.9165(3) Å, V = 60.075(7) Å³ for x = 0, and a = 3.9624(2) Å, V = 62.212(5) Å³ for x = 0.5. The observed expansion for x = 0.5 was expected because of the superior ionic radius of $Ba^{2+}$ vs. $Sr^{2+}$.

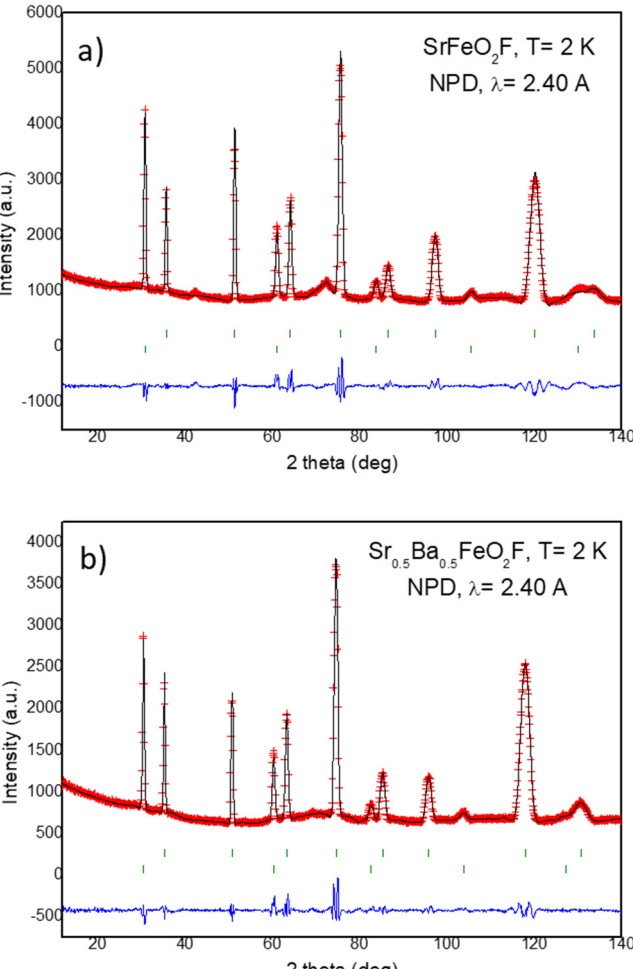

**Figure 3.** Experimental points (red crosses), calculated profile (black line), difference (blue line) and Bragg reflections (green symbols) NPD patterns at 2 K for (**a**) $SrFeO_2F$ and (**b**) $Sr_{0.5}Ba_{0.5}FeO_2F$.

The patterns did not show any additional reflections other than those coming from the G-type antiferromagnetic structure, defined with a propagation vector k = ($^1/_2$, $^1/_2$, $^1/_2$). This is characterized by a perfect antiferromagnetic coupling between Fe spins along the three crystallographic directions. No canting was detected from our NPD data. The propagation vector k = ($^1/_2$, $^1/_2$, $^1/_2$) implies that there are intense additional peaks in the NPD patterns. These magnetic peaks correspond to the second series of tick marks present in Figure 3a,b.

The magnetic structure was thus modeled with collinear Fe spins directed along the *c* axis, as shown in Figure 4a. At 2 K, the magnetic arrangement is fully developed, with refined magnetic moments for Fe of 4.14(3) μB/Fe for x = 0 and 4.25(5) μB/Fe for x = 0.5 at the lowest temperature of 2 K. The evolution of the ordered magnetic moments in the studied temperature range 2–160 K is very subtle, since the antiferromagnetic ordering temperature, $T_N$, in much superior to RT (Figure 4b). The variation in unit-cell parameters in this T interval is represented in Figure 4c, where the expected unit-cell expansion is observed.

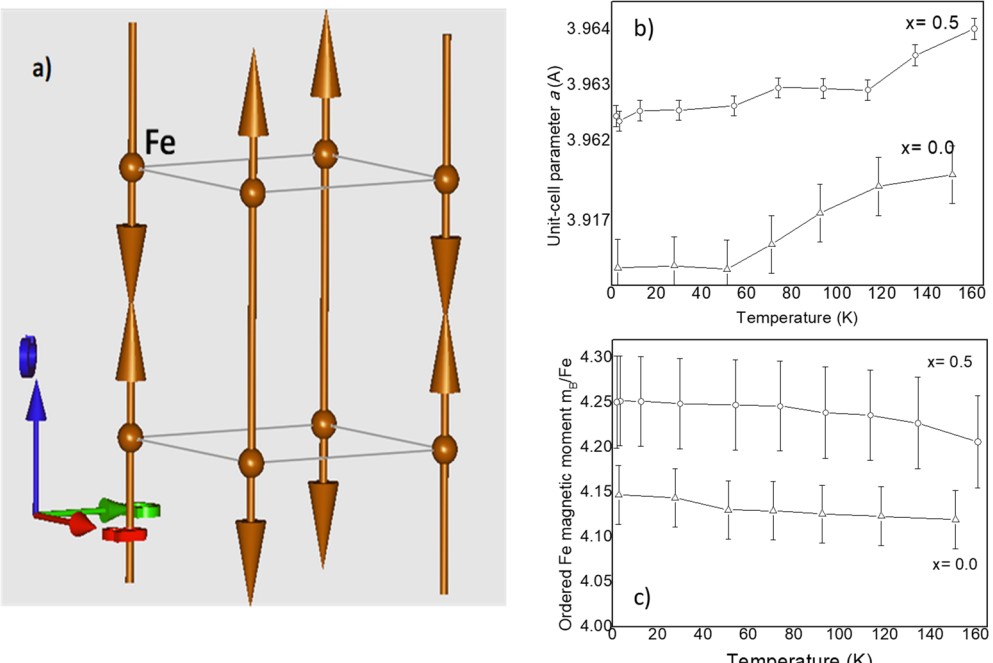

**Figure 4.** (**a**) View of the G-type magnetic structure, with AFM coupling of collinear Fe spins directed along the c axis; (**b**) thermal variation of the ordered magnetic moments for Fe for x = 0.0 and 0.50; (**c**) thermal evolution of the unit-cell parameters for x = 0.0 and 0.50.

*Mössbauer Analysis*

The Mössbauer spectra of the perovskites with x = 0 ($SrFeO_2F$) and x = 0.5 ($Sr_{0.5}Ba_{0.5}FeO_2F$) are illustrated in Figure 5a,b. The spectra at 300 K and 77 K, respectively, can be deconvoluted into two doublets and three sextets. From the FIR of the spectra, the hyperfine magnetic field can be obtained. It decreases upon temperature raising from 77 K to 723 K (not shown). The magnetic fields from the three sextets are 56.19 T, 54.11 T, and 52.24 T, the ratio $\mu_g/\mu_e$ (magnetic moment ground state/excited state) = −1.75098606, −1.75075075, and −1.75113122, and using the expression $D16 = c.H_{hyp}(\mu_g - 3\mu_e)/E_\gamma$ where c is the speed of light in vacuum (m/s), $H_{hyp}$ in T, and $D16$ in mm/s (distance between the first and sixth peak) $E_\gamma = 14.413$ keV, we obtained the values for the magnetic moments at 77 K and 300 K.

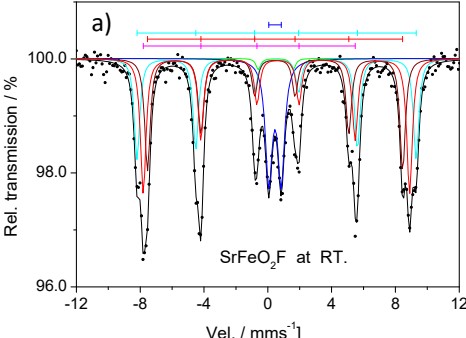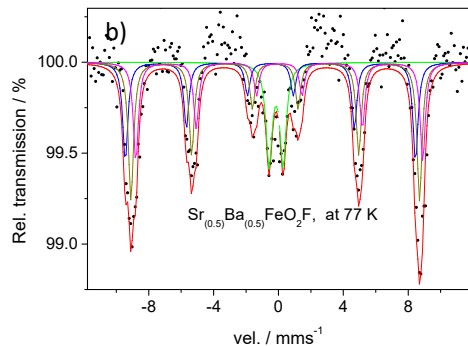

**Figure 5.** Mössbauer spectrum of perovskites $SrFeO_2F$ at (**a**) 300 K (RT) and (**b**) $Sr_{0.5}Ba_{0.5}FeO_2F$ at 77 K.

The spectrum collected at 773 K only displays two doublets, demonstrating that the magnetic transition is placed below this temperature. The isomer shifts (IS) are symptomatic of the $Fe^{3+}$ presence. IS also varies with temperature, as expected from the second-order Doppler shift. Plotting the variation of the hyperfine magnetic field with the temperature, we see that is very similar to the one calculated by the magnetic field expression (7), where Bo stands for the magnetic hyperfine field at 0 K (Bo = 57 T) and $T_N$ is the magnetic ordering temperature. In our calculation, B = 55.5 T at 77 K, B = 1.9 T at 300 K, B = 39.1 T at 573 K, and B = 19.4 T at 723 K. Using these values in the previous formula, we obtain $T_N$ = 739.8 K and Bo = 57.6 T. The value of $T_N$ is higher than that previously reported ($T_N$ = 685 K [14,26]) and the one determined by temperature-dependent NPD [8] located somewhere between 698 K and 723 K. This discrepancy can be related to a slightly different content of F (associated with the presence of $SrF_2$ in the NPD patterns [8]) and a concomitantly higher oxidation state of Fe. In our measurements at 723 K, we are still not in the paramagnetic region, in contrast to the spectrum previously reported at 700 K in [22], which already corresponds to a paramagnetic state. This is the case in our 773 K spectrum, where two doublets are observed, corresponding to the paramagnetic phase. At 923 K, there is only one doublet with a relative area of 29.90% and a singlet with a relative area of 70.08%, both characteristic of $Fe^{3+}$. The Mössbauer spectra for $Sr_{0.5}Ba_{0.5}FeO_2F$ at 77 K and 300 K show three magnetic sextets and two doublets. They are symptomatic of the presence of $Fe^{3+}$ (Figure 5, Table 2). The hyperfine magnetic fields of the three sextets decreases when the temperature increases to 673 K, taking the values of 27.64 T, 25.19 T, and 23.04 T. The 773 K spectrum displays two doublets, indicative of the paramagnetic state with IS 0.19 mms$^{-1}$ and QS of 0.85 mms$^{-1}$ for the first doublet and the second doublet with IS 0.15 mms$^{-1}$ and QS: 1.00 mms$^{-1}$. Both doublets are also characteristic of $Fe^{3+}$. The spectrum at 873 K is a singlet with IS = 0.26 mms$^{-1}$. Using the Mössbauer data, the hyperfine magnetic field at 0 K is $B_0$ = 57.85 T and the Néel temperature, $T_N$ = 716.31 K, in contrast to literature claiming that the onset of antiferromagnetic ordering occurs at $T_N$~670($\pm$10) K [15].

## 4. Discussion

The parameter of proportionality β commented on before in expression (4a) was evaluated using the results of MS data obtained:

$$\beta = \frac{1.38 \times 10^{-23} \times \left(3.955 \times 10^{-10}\right)^3 \times 739.5}{(1/2)\left(3.63 \times 9.274 \times 10^{-24}\right)^2} = 2 \times 5.5744 \times 10^{-4},$$

where β is equivalent to β = 1.31 and $\frac{\mu m}{4\pi}$ $\mu_m$ stands for the magnetic permeability of medium.

The calculation of expression (4) was performed using the program Mathematica taking for N different values from N = 1 to N = 50:

$$\xi\,(N = 1)/\left(\frac{\mu_m}{4\pi}\,\frac{\mu_1\,\mu_2}{a^3}\right) = -1.29289;$$

$$\xi\,(N=2)\Big/\left(\frac{\mu_m}{4\pi}\frac{\mu_1\,\mu_2}{a^3}\right)=-1.31228;$$

$$\xi\,(N=3)\Big/\left(\frac{\mu_m}{4\pi}\frac{\mu_1\,\mu_2}{a^3}\right)=-1.3190;$$

$$\xi\,(N=4)\Big/\left(\frac{\mu_m}{4\pi}\frac{\mu_1\,\mu_2}{a^3}\right)=-1.3219$$

$$\xi\,(N=50)\Big/\left(\frac{\mu_m}{4\pi}\frac{\mu_1\,\mu_2}{a^3}\right)=-1.32294$$

We see that when we increase the number of atoms near the central atom, in this case to N = 50, the interaction energy stabilizes up to a certain value, and in this case β takes a value of 1.3219.

In a previous work [1], the dependence of the Néel temperature with the lattice parameter—a (1) was expressed as:

$$T_N(x) = T_{No}.\exp\left(-k.x^2\right), \tag{5}$$

where k is a constant with a value of 0.13775, which is the slope of curve $\log_e T_N(x)''$ vs. $x^2$.

If we use the constant $k_o$ related to the lattice parameters of perovskite $Sr_{1-x}Ba_xFeO_2F$, $a_x$, and its concentration, $x$, the content of (Ba/Sr): $k_o = \frac{\Delta a}{\Delta x} = \frac{a_{Ba}-a_{Sr}}{1-0} = 0.1060$. In expression (5), the constant k can be compared to $k_o$. If we use logarithm of base b = 3.6544, we get the following relation (Figure 6a,b): $\log_b\left(T_N/T_{No}\right) = -k_o.x^2$.

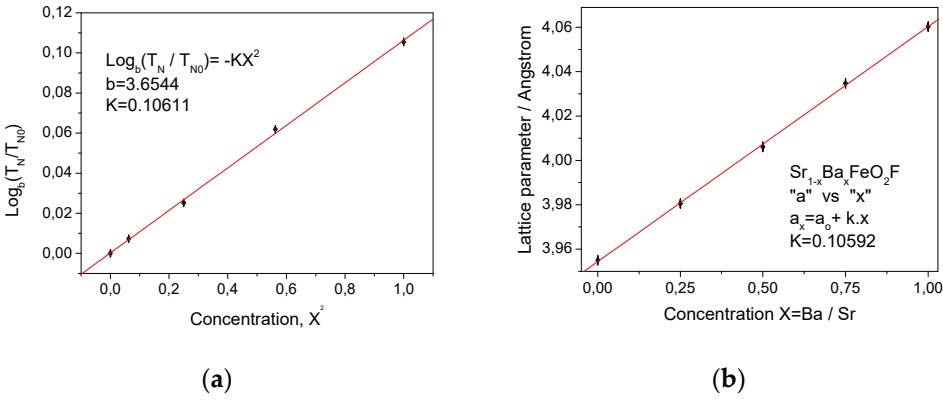

**Figure 6.** (a) Dependence of $\log_b\left(T_{NX}/T_{N0}\right)$ on squared concentration $x^2$; (b) dependence of the lattice parameter $a_x$ on concentration $x$. In both cases, the constant of proportionality is quite similar: k = 0.10611 for (a) ≈ $k_o$ = 0.10592 for (b).

Then $T_{Nx} = T_{No}b^{-k_o.x^2}$. Making some changes, the expression takes the form, $T_{Nx} = T_{No}.b^{-(a_x-a_{Sr})^2/k_o}$ or:

$$T_{Nx} = T_{No}\,b^{-[(a_x-\,a_{Sr})^2/(a_{Ba}-\,a_{Sr})]} \tag{6}$$

where $a_x$ and $a_{Sr}$ are the lattice parameters of the perovskites, and $T_{Nx}$ and $T_{N0}$ are the Néel temperature of $Sr_{1-x}Ba_xFeO_2F$ and $SrFeO_2F$ perovskites, respectively. We see that $\log_b\left(T_N/T_{No}\right)$ has a linear dependence on $x^2$ with a constant of proportionality $k_o'$, which is equivalent to $k_o$ = 0.106011. This is the same proportionality constant $k_o$ of the lattice parameter <a> as a function of concentration $x - k_o = \frac{\Delta a}{\Delta x} = \frac{a_{Ba}-a_{Sr}}{1-0} = 0.10509$.

The thermal evolution of the magnetic hyperfine field for this perovskite series may be articulated as:

$$B = B_o(1 - (T/T_{N0}\exp[-0.13775x^2])^\alpha$$

with α = 0.273 and $T_{N0}$ stands for the temperature of antiferromagnetic ordering of the perovskite with x = 0 ($SrFeO_2F$).

$B_{hyp}$ is the magnetic hyperfine field, given by:

$$B = B_o \ (1 - T/T_N)^{\alpha} \tag{7}$$

where $B_o$ is the magnetic hyperfine field at 0 K, ($B_o$ = 57 T) and the $\alpha$ parameter usually varies in the range $0.25 < \alpha < 0.33$. From our spectrum, we obtain $\alpha = 0.2736$, for the magnetic order temperature $T_N$ = 740.8 K, for $x$ = 0.0 and the hyperfine magnetic field (at 0 K) $B_o$ = 57.6 T. (See Tables 1–3).

**Table 1.** Structural parameters and reliability factors for $Sr_{1-x}Ba_xFeO_{3-y}F_y$, obtained from NPD data at room temperature (300 K) and the relations $(T_N/T_{N0})$, $log_b (T_N/T_{N0})$.

| $x$ (Ba Content) | 0 | 0.25 | 0.50 | 0.75 |
|---|---|---|---|---|
| $a$ (Å) | 3.95500(7) | 3.98055(6) | 4.00610(5) | 4.03476(6) |
| V (Å$^3$) | 61.864(2) | 63.071(2) | 64.293(1) | 65.683(2) |
| $T_N$ (K) | 740.08 | 733.10 | 715.31 | 683.40 |
| $T_N/T_{No}$ | 1 | 0.99056 | 0.96788 | 0.922873 |
| $Log_b (T_N/T_{No})$, b = 3.6544 | 0 | −0.0073122 | 0.0251911 | −0.061920 |
| Sr/Ba 1$b$ ($^1$/$_2$ $^1$/$_2$ $^1$/$_2$) | | | | |
| B (Å$^2$) | 0.80(4) | 0.81(3) | 0.87(3) | 0.84(3) |
| Fe 1$a$ (0 0 0) | | | | |
| B (Å$^2$) | 1.62(4) | 1.89(3) | 2.29(3) | 2.68(3) |
| $\mu_B$ | 3.63(4) | 3.50(3) | 3.37(3) | 3.40(2) |
| O/F 3$d$ (0 0 $^1$/$_2$) | | | | |
| B (Å$^2$) | 2.36(4) | 1.55(3) | 1.30(3) | 1.10(2) |
| Occupancy O/F | 1 | 1 | 0.98(1) | 0.96(1) |
| Main bond distances (Å) | | | | |
| Sr-O/F (×12) | 2.79661(4) | 2.81467(3) | 2.83274(2) | 2.85301(3) |
| Fe-O/F (×6) | 1.97750(4) | 1.99028(3) | 2.00305(2) | 2.01738(3) |

**Table 2.** Position of the peaks of the Mossbauer spectrum of perovskites $SrFeO_2F$ and $Sr_{0.5}Ba_{0.5}FeO_2F$ (marked as A-first peak, B, C, D, E, F-sixth peak), the magnetic hyperfine field and the magnetic moment for $SrFeO_2F$ and $Sr_{0.5}Ba_{0.5}FeO_2F$.

| Peaks | A | B | C | D | E | F | $SrFeO_2F$ | $H_{hyp}$/T | T/K |
|---|---|---|---|---|---|---|---|---|---|
| Sextet | −8.685 | −4.981 | −1.178 | 1.678 | 5.481 | 9.385 | [mm/s] | 56.19 | |
| | μ = 0.093104 | | [mm/Ts] = 4.475963 | | [neV/T] = 3.69182 | | Nuclear Bohr magn. | | 77 |
| Sextet | −8.205 | −4.414 | −0.783 | 1.943 | 5.574 | 9.045 | [mm/s] | 53.64 | 300 |
| | μ = 0.093102 | | [mm/Ts] = 4.475886 | | [neV/T] = 3.691757 | | Nuclear Bohr magn. | | |
| | | | | | | | $Sr_{0.5}Ba_{0.5}FeO_2F$ | | |
| Sextet | −9.424 | −5.551 | −1.678 | 1.23 | 5.13 | 9.976 | [mm/s] | 57.22 | |
| | μ = 0.097348 | | [mm/Ts] = 4.680003 | | [neV/T] = 3.860115 | | Nuclear Bohr magn. | | 77 |
| Sextet | −8.18 | −4.36 | −0.739 | 1.979 | 5.6 | 9.02 | [mm/s] | 53.48 | T/K |
| | μ = 0.09311 | | [mm/Ts] = 4.476243 | | [neV/T] = 3.692051 | | Nuclear Bohr magn. | | 300 |

**Table 3.** The Predicted by formula (9) and Observed magnetic hyperfine field B ($x$, T) depending of concentration ($x$) and temperature (T/K) for the series of perovskites $Sr_{1-x}Ba_xFeO_2F$.

| $B(x, T) = 56.0\left(1 - \frac{T}{740.08}b^{0.10611.x^2}\right)^{\frac{1}{b}}$, b = 3.6544 | | | | | | | |
|---|---|---|---|---|---|---|---|
| x\T[K] | 77 | 300 | 473 | 573 | 673 | 723 | 740.08 |
| 0.00 | 54.34 \| 54.11 | 48.57 \| 51.62 | 42.37 \| - | 37.26 \| 39.18 | 29.03 \| - | 19.96 \| - | 0 \| 0 |
| 0.25 | 54.32 \| 56.00 | 48.49 \| 51.62 | 42.19 \| - | 36.96 \| 38.87 | 28.32 \| - | 17.63 \| 15.59 | - |
| 0.50 | 54.28 \| 55.35 | 48.25 \| 51.00 | 41.63 \| 44.47 | 35.98 \| 37.94 | 25.79 \| 25.18 | - | - |
| 0.75 | 54.20 \| 56.59 | 47.83 \| 50.53 | 40.62 \| - | 34.11 \| 37.31 | 18.51 \| - | - | - |
| 1.00 | 54.08 \| - | 47.18 \| - | 39.00 \| - | 30.73 \| - | - | - | - |

A further analysis of the variation of Neel's temperature $T_N$ from the Ba contents $x$ shows that the temperature has an exponential dependence on $<x^2>$ (Figure 7). $Log_b = -KoX^2$ , where $K_o = 0.10611$, which is, in fact, the constant of proportionality of the lattice parameter of the perovskite $Sr_{1-x}Ba_xFeO_2F$, then

$$T_{Nx} = T_{No}b^{-koX^2}. \tag{8}$$

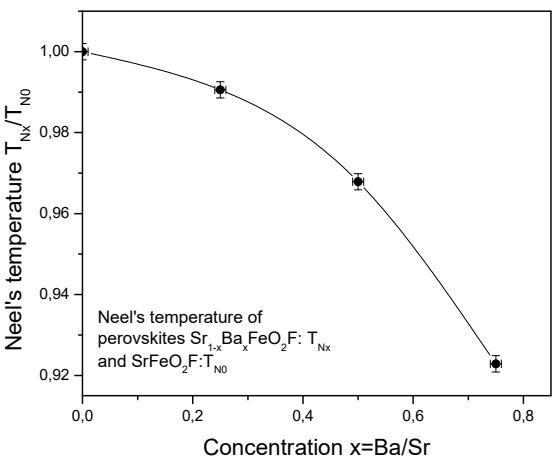

**Figure 7.** Dependence of the relation ($T_{Nx}/T_{No}$) concerning the Néel temperature of perovskites $Sr_{1-x}Ba_xFeO_2F$ and $SrFeO_2F$ vs. concentration $x$.

The variation of the magnetic field with the increase of temperature for all the series of this perovskites model may be expressed as $B(x,T) = B\left(1 - \frac{T}{T_{Nx}}\right)^{\alpha}$, where $T_{Nx}$ is the Neel's temperature of the perovskite with concentration $x$, then the expression for the variation of hyperfine magnetic field from temperature and its concentration, $B(x, T)$ takes the form:

$$B(x, T) = B_0\left[1 - \frac{T}{T_{N0}}\propto^{-kX^2}\right]^{\alpha}$$

With $\alpha = 0.2736$ and the hyperfine field Bo = 57(T) at 0 K, $T_{No}$ is the magnetic order temperature of $SrFeO_2F$ perovskite with $x = 0$.

The magnetic order temperature or Neel's temperature depends on $x$ (the Ba content); this dependence may be expressed as:

$T(x) = T_{N0}\,b^{-kox^2}$ due to expression (3), yields $T(x) = T_{N0}\,b^{\frac{-(a_x-a_{sr})^2}{ko}}$, or to the expression

$T_N(x) = T_{N0}\,b^{\frac{-(a_x-a_{sr})^2}{(a_{Ba}-a_{sr})}}$, where $k_o = a_{Ba} - a_{Sr}$

This expression may be transformed to $T_N(x) = T_{No} \sqrt{\frac{1-tanhyp(ko.\ x^2.Logb)}{1+Tanhyp(ko.x^2\ .Logb)}}$, or convert-

ing the value of <ko. $Log_e$ b = 0.13775> it yields: $T_N(x) = T_{No} \sqrt{\frac{1-tanhyp(0.13775\ x^2)}{1+Tanhyp(0.13775x^2)}}$

The expression for the dependence magnetic hyperfine field from Temperature and concentration x takes the form:

$$B(x, T) = B_0 \left[ 1 - \frac{T}{T_{N0}} b^{kx^2} \right]^{1/b} \tag{9}$$

## 5. Conclusions

The Mössbauer spectra clearly indicate the presence of $Fe^{3+}$ for the four oxides. As expected, the temperature of antiferromagnetic ordering decreases as *x* (the Ba/Sr contents) rises, since the super-exchange interactions between $Fe^{3+}$ neighbors are weakened upon $Ba^{2+}$ incorporation into the lattice as the unit-cell size increases. The magnetic moment obtained by Mössbauer data is lower than those obtained by the NPD. This discrepancy is due to the fact that the Mössbauer data give the magnetic moment of the nucleus and the NPD gives the magnetic moment of the entire atom. Mössbauer data showed that the magnetic hyperfine field decreased upon temperature increase, according to $B(T) = B_o(1 - T/T_{Nx})^{1/b}$, where $B_0$ is the magnetic hyperfine field at 0 K and $T_{Nx}$ is the magnetic order temperature for perovskite with contents *x*, 1/b = 0.2736—parameter.

The magnetic order temperature depends on *x* and temperature T. This dependence may be expressed as:

$$B(x, T) = B_0 \left[ 1 - \frac{T}{T_{N0}} b^{kx^2} \right]^{1/b}$$

**Author Contributions:** J.A.A. carried out the synthesis of the samples and initial structural characterization. M.T.F.-D. collected the neutron data. Data processing and analysis were done by K.K. and C.A.G.-R. The research design and the manuscript were written by K.K., C.A.G.-R. and J.A.A. All the authors discussed the results and commented on the manuscript. All authors have read and agreed to the published version of the manuscript.

**Funding:** This research was funded by the Spanish Ministry of Science and Innovation for the project (PID2021-122477OB-I00).

**Informed Consent Statement:** Not applicable.

**Data Availability Statement:** Experimental data are available upon request.

**Acknowledgments:** We acknowledge the financial support of the Spanish Ministry of Science and Innovation for the project (PID2021-122477OB-I00). K.K. acknowledges the Bulgarian National Science Fund for grant KP-06-N48/5. We are grateful to ILL (France) for making all facilities available.

**Conflicts of Interest:** The authors declare no competing financial or nonfinancial interests.

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
