# Peer review of "Antiferromagnetism and Structure of Sr1−xBaxFeO2F Oxyfluoride Perovskites"

_magnetochemistry, doi:10.3390/magnetochemistry9030078_

Round 1
Reviewer 1 Report
The work makes a very good impression. The choice of topic, research methods is clearly justified. The results are new and interesting, they are well presented and illustrated. The conclusions are based on the obtained results. In order for the work to be accepted for publication, it is necessary to eliminate some editorial errors. Authors should carefully check all text and make appropriate corrections, for example: Abstract, line 11: Sr1-xBaxFeO2F, line 21: SrFeO2F (x=1.0).
Besides, it should be noted that the problem of the formation of the physical properties of oxyfluoride perovskites is one of the most important in condensed matter physics and a significant number of publications are devoted to it. In this regard, I would nice to see in this article more links to recent publications, e.g.,
Hai-Chen Wang, …, A high-throughput study of oxynitride, oxyfluoride and nitrofluoride perovskites J. Mater. Chem. A, 2021, 8501,
Akira Chikamatsu, …, Electronic properties of perovskite strontium chromium oxyfluoride epitaxial thin films fabricated via low-temperature topotactic reaction
Phys. Rev. Materials 4 (2020) 025004,
Reviewer 2 Report
The authors have conducted a research about some properties of oxyfluorides Sr1-xBaxFeO2F and arranged a manuscript entitled "Antiferromagnetism and structure of Sr1-xBaxFeO2F oxyfluoride 2 perovskites". This manuscript can be considered for acceptance after considering the following points:
Line 62: There is subsection 1.2 but subsection 1.1 is not found. The referee suggests to include subsection 1.2 into a new section, for instance, "Research Methodology", alongside the Experimental section
Line 65-76: There is inconsistency in writing quantities, such as lattice parameter and content. It was written with italic "a" but at the other places, it was written with "<a>". The referee suggests to check the consistency of symbols and adapt them with that are commonly used in textbooks or papers.
Abstract and conclusion section: The conclusion seems not well arranged as it is still a result discussion. Please arrange more conclusive paraghraph for the conclusion section which is in line with the abstract. Furthermore, please express the significance of this work based on the results and discussions if any, besides informing the novel synthesis route.
Based on the references list, most of the papers cited in this manuscript are not new. Can you please provide more references in the Introduction that are published up to 4 or 5 years ago?
Figure 4: Please enlarge the font size in the Figure.
